# Biodrug Delivery Systems: Do mRNA Lipid Nanoparticles Come of Age?

**DOI:** 10.3390/ijms24032218

**Published:** 2023-01-22

**Authors:** Matteo Puccetti, Aurelie Schoubben, Stefano Giovagnoli, Maurizio Ricci

**Affiliations:** Department of Pharmaceutical Sciences, University of Perugia, 06123 Perugia, Italy

**Keywords:** RNA lipid nanoparticles, RNA delivery systems, biodrug nanotechnology

## Abstract

As an appealing alternative to treat and prevent diseases ranging from cancer to COVID-19, mRNA has demonstrated significant clinical effects. Nanotechnology facilitates the successful implementation of the systemic delivery of mRNA for safe human consumption. In this manuscript, we provide an overview of current mRNA therapeutic applications and discuss key biological barriers to delivery and recent advances in the development of nonviral systems. The relevant challenges that LNPs face in achieving cost-effective and widespread clinical implementation when delivering mRNA are likewise discussed.

## 1. Introduction

Conventional drug administration routes suffer from low selective distribution, poor solubility, and fast elimination, and they damage adjacent healthy tissues. Thus, drug delivery systems, which can deliver drugs to target sites, have been broadly investigated to increase drug bioavailability and to diminish its side effects. Micro- and nanoparticle formulations are of great interest as materials for the delivery of conventional drugs or biopharmaceuticals. Microparticles can be utilized to prepare longstanding stock formulations that are stable for weeks to months and can be injected through intramuscular and/or subcutaneous methods. Nanocarriers can quickly penetrate cells through intracellular endocytic routes and efficiently release pharmaceuticals into a local microenvironment.

The rational design and application of mRNA-based formulations have recently led to some key successes in treating human diseases. mRNA technology allows for the direct production of proteins in vivo, thus avoiding the need for lengthy drug development cycles and complex production workflows. As such, mRNA formulations can significantly improve the biological therapies that have been used thus far. Despite its many advantages, mRNA is inherently fragile and has specific delivery requirements. By leveraging the engineering flexibility of nanobiotechnology, mRNA payloads can be incorporated into nanoformulations such that they do not result in unwanted immune responses, are targeted to tissues of interest, and can be delivered to the cytosol, resulting in improved safety while enhancing bioactivity [1,2].

Synthetic mRNA technologies represent a versatile platform that can be used to develop advanced drug products. The remarkable speed with which vaccine development programs were able to design and manufacture safe and effective COVID-19 vaccines has rekindled interest in mRNA technology, particularly for future pandemic preparedness. Although recent research and development approaches have largely focused on advancing mRNA vaccines and large-scale manufacturing capabilities, the technology has also been used to develop various immunotherapies, gene editing strategies, and protein replacement therapies. Within the mRNA technology toolbox lie several platforms, design principles, and components that can be adapted to modulate immunogenicity, stability, in situ expression, and delivery [3].

Artificial nucleic acid nanostructures have become an emerging research hotspot as gene carriers with low cytotoxicity and immunogenicity for therapeutic approaches. Recent progress has been made in the design and functional mechanisms of nucleic acid-based artificial nano-vectors, especially for exogenous siRNA and antisense oligonucleotide delivery. As a result, different nanocarriers are being tested in targeted gene editing and in the co-delivery of anticancer drugs and small ribonucleic acids. Prior to the emergence of SARS-CoV-2, the potential use of mRNA vaccines for a rapid pandemic response had been well-described in the scientific literature; however, during the SARS-CoV-2 outbreak, we witnessed the large-scale deployment of the platform in a real pandemic setting. Of the three RNA platforms evaluated in clinical trials ((1) conventional, non-amplifying mRNA; (2) base-modified, non-amplifying mRNA that incorporates chemically modified nucleotides; and (3) self-amplifying RNA (saRNA)), the base-modified, non-amplifying mRNA technology emerged with superior clinical efficacy.

As mentioned above, there are two main types of mRNA technologies: conventional mRNA and saRNA (reviewed in [4]). The basic components of an RNA transcript are a 5′ cap and 5′ untranslated region (UTR), an open reading frame (ORF), a 3′ UTR, and a polyadenosine tail, all of which are essential for in situ protein translation. saRNAs require additional components in the form of sequences encoding four non-structural proteins (nsP1–4) derived from Alphavirus genomes, which form the RNA-dependent RNA polymerase complex, and the 5′ and 3′ conserved sequence elements, both of which are required for the self-propagation of RNA. The sequence encoding the gene of interest is typically included downstream of nsP1–4 under the control of a subgenomic promoter.

Synthetic mRNA production starts with the design and synthesis of a DNA template that encodes the necessary components of the transcript downstream of a bacteriophage promoter (usually T7, T3 or SP6). The mRNA can then be transcribed from the linear template in a highly efficient and cell-free process known as in vitro transcription. In its simplest form, this requires an RNA polymerase corresponding to the promoter encoded in the template, ribonucleotide triphosphates, and an appropriate buffer. The capping of the mRNA can be achieved during in vitro transcription with co-transcriptional cap analogs. Alternatively, the enzymatic addition of a 5′ cap is possible. Purified mRNA is subsequently formulated in a lipid nanoparticle containing a mixture of cationic or ionizable lipids and excipients. Upon entry into the target cell, the protein is immediately translated using host translational machinery and can function within the cytoplasm, be trafficked to the cell-membrane or nucleus, or secreted, depending on the design of the mRNA. In the case of saRNA, the initial in situ translation of the nsP1–4 proteins ensures the exponential replication and subsequent translation of the subgenomic RNA. A major advantage of this platform is that the in situ translation of mRNA in the cytoplasm of host cells allows for native protein folding and post-transcriptional modifications. As such, therapeutics and antigens are expressed in conformations that often mimic native host proteins and viral epitopes, respectively. Other key attributes, including reduced genotoxicity, flexible modular design principles, and small manufacturing footprints, have helped mRNA platforms overcome the limits of traditional vaccine production pipelines. Importantly, this platform may be considered a stepping-stone for disease-burdened developing countries to gain vaccine manufacturing independence.

So-called saRNA-based technology has also been widely tested in numerous directions apart from infectious diseases, including cancer prevention and the treatment of inherited disorders. Interestingly, saRNA-based technology is believed to display more developed RNA therapy compared with conventional mRNA technique in terms of its lower dosage requirements, relatively fewer side effects, and long-lasting effects. Nevertheless, there are still some challenges that need to be overcome in order to achieve saRNA-based drug approval in clinics [5].

Lipid nanoparticles (LNPs) have emerged as a very promising delivery method. However, when intravenously delivering LNPs, most of the cargo is trapped by the liver. Alternatively, injecting them directly into organs, such as the brain, requires more invasive procedures. Therefore, developing more specific LNPs is crucial for their future clinical use. Modifying the composition of the lipids in LNPs allows for the more specific delivery of the LNPs to some organs (Figure 1). This topic has been extensively dealt with elsewhere [6].

## 2. mRNA Non-Viral Delivery Approaches

Messenger RNAs (mRNAs) present great potential as therapeutics for the treatment and prevention of a wide range of human pathologies, allowing for protein replacement, vaccination, cancer immunotherapy, and genomic engineering. Despite advances in the design of mRNA-based therapeutics, a key aspect for their widespread translation to clinical use is the development of safe and effective delivery strategies. To this end, non-viral delivery systems including peptide-based complexes, lipidic or polymeric nanoparticles, and hybrid formulations are attracting growing interest. Despite displaying somewhat reduced efficacy compared with viral-based systems, non-viral carriers offer important advantages in terms of biosafety and versatility.

In this review, we intend to provide an overview of current mRNA therapeutic applications and discuss key biological barriers to delivery and recent advances in the development of non-viral systems [7]. The classical therapeutic applications of mRNAs are protein replacement strategies, vaccine production, and gene editing (Figure 2).

### 2.1. mRNA Non-Viral Delivery Approaches for Proteins

In protein replacement, an in vitro transcribed mRNA is used to induce the expression of a protein whose malfunction or absence determines a specific pathological condition. mRNA-induced protein expression has also been applied to cell reprogramming with the goal of modulating cell fate and function. The first description of such an approach was described by Yamanaka and Gurdon, winners of the 2012 Nobel Prize. The authors demonstrated that the introduction of OCT4, SOX2, MYC, and KLF4 transcription factors (indicated as a Yamanaka factor mix) in human somatic cells induces their reprogramming in induced pluripotent stem cells. Based on these findings, mRNAs encoding for the mix of transcription factors have been used to produce induced pluripotent stem cells. Similar approaches have been developed for the reprogramming of other cell types (reviewed in [7]). As is discussed later, Phase I/II clinical trials are ongoing for the inhalation of LNPs for the treatment of cystic fibrosis via the protein target CFTR [8]. (The cystic fibrosis transmembrane conductance regulator or CFTR protein helps to maintain the balance of salt and water on many surfaces in the body, such as the surface of the lung. When the protein is not working correctly, chloride—a component of salt—becomes trapped in cells.)

### 2.2. mRNA-Based Vaccines

Traditional viral vaccines rely on the safe and effective administration of whole (inactivated or attenuated) or partial (subunit) viruses as training tools for the immune system. However, these strategies are often slow in their development, making it hard to keep pace with newly emerging virus strains. In contrast, the facile payload interchangeability of mRNA-based nanomedicine platforms can significantly streamline the development process. Optimal vaccine targets can be quickly discovered through genetic sequencing, rapidly yielding templates for subsequent large-scale mRNA production. The rapid discovery process, synergistically paired with relatively inexpensive biomanufacturing costs for LNP formulations, have enabled mRNA vaccine candidates to reach clinical testing and receive regulatory authorization much faster than traditional vaccines. This was exemplified by the recent development and deployment of the Pfizer–BioNTech BNT162b2 and Moderna mRNA-1273 mRNA vaccines to combat the COVID-19 pandemic. Both vaccines contain nucleoside-modified mRNAs that induce the membrane-bound expression of a perfusion-stabilized, full-length severe acute respiratory syndrome coronavirus 2 (SARS-CoV-2) spike protein. In each case, the mRNA vaccines were formulated using LNPs for intramuscular injection. The rapid development and potent efficacy of these vaccines will serve as a strong benchmark for the advancement of future mRNA-based vaccines against a broad set of diseases. Despite the strong successes of these vaccines, the need for frozen storage and short-term usability when thawed represent a barrier for widespread global distribution. Fortunately, strong efforts are being undertaken to overcome these challenges, and additional mRNA-based nano-vaccines against COVID-19 are actively being developed. A representative list of clinical trials in the United States investigating new mRNA-based interventions in viral infections is shown in Table 1; for an exhaustive list, see [9].

In addition, to increase the efficacy of mRNA-based vaccines, additional strategies such as self-amplifying mRNA vaccines are being developed. Self-amplifying mRNA vaccines use an engineered RNA virus genome in which the genes for the antigens of interest are inserted in place of those encoding the virus structural proteins while the genes for the virus RNA replication machinery are kept intact. In contrast to traditional mRNA-based vaccines, self-amplifying mRNA vaccines allow for the intracellular replication of antigen-encoding RNA, resulting in a higher level of antigen production that enhances vaccine efficacy. Since the virus structural protein genes are removed, it is not possible to produce infectious virions, thereby securing a safe vaccine profile. However, self-amplifying mRNA vaccines show some difficulties compared with mRNA vaccines. They have a necessarily higher molecular size due to the presence of the viral-derived genes for the RNA replication machinery, which can also cause immunogenicity, thus limiting their potential repeated use (refer to [10] for further detail). Thus far, the self-amplifying mRNA vaccine platform has been applied against diverse viruses including influenza, Ebola, hepatitis C, rabies virus, *Toxoplasma gondii*, human cytomegalovirus, and HIV-1 [7].

Cancer immunotherapy is another promising application of mRNA vaccines [7]. To date, two different strategies have been utilized for mRNA-based cancer immunotherapy. The first utilizes mRNA encoding for tumor-associated antigens [11]. The mRNA is transfected ex vivo in antigen-presenting cells such as dendritic cells to activate cytotoxic T cells [12]. A second strategy for mRNA-based cancer immunotherapy involves the generation of modified T cells expressing chimeric antigen receptors (CARs) derived from the fusion of protein fragments binding to specific tumor epitopes and selected T-cell receptor domains. The procedure requires the ex vivo genetic modification of patient-derived T cells (CAR T cells) with CAR mRNA via viral/non-viral gene transfer or via the direct transfer of in vitro transcription mRNA. This is followed by cell expansion and re-infusion to the patient. Infused CAR T cells determine anti-cancer immune responses, targeting cancer cells and tumor regression [13].

### 2.3. mRNA for Gene Editing

In addition to protein replacement and vaccines, more recently, the development of CRISPR (Clustered Regularly Interspaced Short Palindromic Repeat) technology led to the application of mRNAs in gene editing and extended their use in pathologies requiring not only protein expression but also gene knockout [7].

The basic CRISPR system was discovered as a bacterial mechanism of defense requiring the expression of the mRNA encoding for the nuclease CRISPR-associated protein 9 (Cas9) and that of a short guide RNA (gRNA) that directs the nuclease to a specific DNA location that is then cleaved [14]. Traditionally, a DNA plasmid containing both the *Cas9* gene and the gRNA is used. This system is limited by potential unwanted cuts within the genome due to the persistent expression of Cas9. Hence, the use of mRNA coding for Cas9 or other nucleases has been proposed as a valid alternative to reduce off-target effects thanks to the transient nature of mRNA [15].

Based on CRISPR/Cas9 technology, Gillmore and colleagues developed a gene editing therapeutic agent named NTLA-2001 that could be the first curative treatment for ATTR (transthyretin) amyloidosis. This drug takes advantage of lipid nanoparticles designed to deliver a two-part genome editing system, comprising a human-optimized mRNA molecule encoding for the Cas9 protein and a single guide RNA molecule specifically targeting the human gene encoding for transthyretin (TTR), to the liver. According to data obtained by authors in preclinical in vitro and in vivo studies, NTLA-2001 may be able to produce the efficient and durable knockdown of TTR expression with a single administration. The safety of the treatment was found to be very satisfactory, since only a few and very mild adverse effects occurred in analyzed patients [16].

## 3. Lipid Nanoparticle-Mediated Delivery of Oligonucleotides and mRNA Therapeutics

The optimal delivery system for RNA-based therapies should allow for a simple and non-invasive route of delivery to the body, avoid unwanted immune responses, allow for penetration through tissues and biological barriers (e.g., blood–brain barrier, BBB), provide cell and tissue specificity, provide stability in extra- and intracellular environments, engage in fast cell penetration, be efficiently localized to subcellular compartments, and allow for the targeting oligonucleotide to bind specifically (on-target/off-target ratio) and stably to the intended target including full compatibility with the cellular machinery and very limited interference with non-targeted cellular processes. The covalent conjugation of functional molecules to RNA, small molecule conjugation, carbohydrate conjugation, hydrophobic molecule conjugation, peptide conjugation, and aptamer conjugation have all been tested as RNA carriers [17].

Lipid-based nanoparticles are among the most well-established nanoparticle systems applied in the delivery of RNA. By encapsulating RNA, LNPs protect cargo from extracellular nucleases, thus allowing for the safe delivery of unmodified oligonucleotides to cells. The complexation of oligonucleotides with cationic lipids, such as 1,2-di-*O*-octadecenyl-3-trimethylammonium-propane (DOTMA) and 1,2-dioleoyl-3-trimethylammonium-propane (DOTAP), neutralizes charges, condenses long oligonucleotide chains, and improves encapsulation when mixed with zwitterionic lipids, such as 1,2-dioleoyl-sn-3-phosphoethanolamine (DOPE) and dioleoyl phosphatidylcholine (DOPC). The delivery of oligonucleotides only complexed by DOTMA or other cationic lipids is known as lipofection, and it is the most popular in vitro transformation method. Unfortunately, most lipofection reagents are not suitable for in vivo delivery due to their high cytotoxicity, thus making liposomal formulations more suitable for therapeutic applications.

LNPs are now mostly formulated with the following four lipid components: (i) helper lipid to encapsulate cargo, (ii) ionizable lipid to enhance endosomal escape and delivery, (iii) cholesterol to promote stability, and (iv) lipid-anchored poly(ethylene glycol) (PEG) to reduce immune system recognition and improve biodistribution. Other targeting moieties, such as antibodies and peptides, may be added to further direct localization. Lipid components are often combined with mRNA via microfluidic mixing to form LNPs [18].

A new methodology that enables the controllable delivery of nucleic acids to target tissues, termed selective organ targeting (SORT), was recently developed. SORT LNPs involve the inclusion of SORT molecules that were found to accurately tune delivery to the liver, lungs, and spleen of mice after intravenous administration. Nanoparticles can be engineered to target specific cells and organs in the body via passive, active and endogenous targeting mechanisms that require distinct design criteria. SORT LNPs are modular and can be prepared using scalable, synthetic chemistry and established engineering formulation methods [19].

Nonetheless, LNPs still face several delivery barriers, including nonspecific serum protein interactions, rapid clearance, off-target localization, and degradation in the endosome. Furthermore, mRNA delivery induces transient protein production, requiring repeated administration for sustained expression. Finally, the development of anti-PEG antibodies raises concerns about potential allergic responses to LNPs; Pfizer–BioNTech BNT162b2 and Moderna mRNA-1273 mRNA vaccines against COVID-19, viral mRNA, or oligonucleotides thereof may exert dual effects—including potent proinflammatory activity or, vice versa, immune suppression—depending on dosage and predominant target cell type due to direct interactions with Toll-like receptors (TLRs), a class of proteins that play a key role in the innate immune system [20].

## 4. Applications

Post-transcriptional gene-silencing targets and degrades mRNA transcripts, silencing the expression of specific genes. RNA interference technology, using synthetic and structurally well-defined short double-stranded RNA (small interfering RNA (siRNA)), has rapidly advanced in recent years [21].

Patisirean, sold under the brand name Onpattro, is a drug used for the treatment of polyneuropathy in people with hereditary transthyretin amyloidosis, a fatal rare disease that is estimated to affect 50,000 people worldwide. Onpattro, an RNA interference drug, was the first FDA-approved nucleic acid LNP therapeutic for the treatment of polyneuropathy caused by transthyretin amyloidosis. It is a gene-silencing drug [22] that interferes with the production of an abnormal form of transthyretin. Patisiran utilizes a novel approach to target and reduce the production of the transthyretin protein in the liver via the RNA interference pathway. The siRNA-active component of Patisiran is formulated into lipid nanoparticles that protect the RNA and facilitate its delivery to target tissues. The lipid nanoparticle formulation includes buffer components, as well as the lipid components DLin-MC3-DMA, distearoylphosphatidylcholine, cholesterol, and the PEGylated slipid DMG-PEG 2000 [23,24]. A CRISPR–Cas9 in vivo gene editing approach for transthyretin amyloidosis is also being investigated in a clinical trial (funded by Intellia Therapeutics and Regeneron Pharmaceuticals; ClinicalTrials.gov number, NCT04601051) [16].

Similarly, hepatic fibrosis secondary to hepatitis C virus infection can lead to cirrhosis and hepatic decompensation. Sustained virologic response is possible with direct-acting antiviral drug regimens; however, patients with advanced fibrosis have an increased risk for hepatocellular carcinoma. Heat shock protein 47 (HSP47), a key collagen chaperone, has been implicated in fibrosis development. Data are available on the efficacy and safety of BMS-986263, a lipid nanoparticle delivering small interfering RNA designed to degrade HSP47 mRNA, for the treatment of advanced fibrosis [25]. NCT03420768 is a randomized (1:1:2), Phase 2 placebo-controlled trial being conducted at a hepatology clinic in the United States. Patients with HCV-SVR (for ≥1 year) and advanced fibrosis received once-weekly i.v. infusions of placebo or BMS-986263 (45 or 90 mg) for 12 weeks.

In a first-in-human study, immune responses to a rabies virus glycoprotein (RABV-G)-mRNA vaccine were found to be dependent on the route of administration, necessitating specialized devices. Following successful preclinical studies with mRNA encapsulated in lipid nanoparticles (LNPs), an mRNA–LNP formulation (CV7202) is currently being investigated in a clinical trial (ClinicalTrials.gov Identifier: NCT03713086).

Pfizer–BioNTech and Moderna mRNA–LNP coronavirus disease 2019 (COVID-19) vaccines were given FDA emergency-use authorization in 2020 [26].

Preclinical and clinical studies are ongoing for the inhalation of LNPs for the treatment of cystic fibrosis via the protein target CFTR [8,27,28,29].

Despite major successes (e.g., mRNA vaccines developed against SARS-CoV-2 to control the COVID-19 pandemic), the development of therapies for other diseases is still limited by the inefficient delivery of oligonucleotides to specific tissues and organs and the often prohibitive costs for the final drug. This is even more critical when targeting multifactorial disorders and patient-specific biological variations [17].

As an example, in cystic fibrosis, despite lipid LNPs’ success in the effective and safe delivery of mRNA vaccines, the development of an inhalation-based mRNA therapy for lung diseases remains challenging. LNPs tend to disintegrate due to shear stress during aerosolization, leading to ineffective delivery. Therefore, LNPs need to remain stable through the process of nebulization and mucus penetration yet labile enough for endosomal escape. To meet these opposing needs, PEG lipids can be used to enhance the surficial stability of LNPs with the inclusion of a cholesterol analog, β-sitosterol, to improve endosomal escape. Increased PEG concentrations in LNPs was found to enhance shear resistance and mucus penetration, and β-sitosterol was shown to provide LNPs with a polyhedral shape, facilitating endosomal escape. The optimized LNPs exhibited a uniform particle distribution, a polyhedral morphology, and a rapid mucosal diffusion with enhanced gene transfection. Inhaled LNPs led to localized protein production in mouse lungs without pulmonary or systemic toxicity. The repeated administration of these LNPs led to sustained protein production in the lungs. Lastly, mRNA encoding the cystic fibrosis transmembrane conductance regulator (CFTR) was delivered after nebulization to a CFTR-deficient animal model, resulting in the pulmonary expression of this therapeutic protein. This study demonstrated a rational design approach for the clinical translation of inhalable LNP-based mRNA therapies [8].

Interestingly, while all the siRNA drugs on the market target the liver, the lungs offer a variety of currently undruggable targets that could be treated with RNA therapeutics. Hence, the local pulmonary delivery of RNA nanoparticles could finally enable delivery beyond the liver. Targeted drug delivery technologies potentiate the overall therapeutic efficacy of an indole derivative in a mouse cystic fibrosis setting [30,31]. The administration of RNA drugs via dry powder inhalers offers many advantages related to the physical, chemical, and microbial stability of RNA and nanosuspensions. A recent study was therefore designed to test the feasibility of engineering spray-dried LNP powders. Spray drying was performed using a 5% lactose solution (m/V), and the targets were set to obtain nanoparticle sizes after the redispersion of spray-dried powders of around 150 nm, a residual moisture level below 5%, and RNA losses of below 15% at a maintained RNA bioactivity level. The LNPs consisted of an ionizable cationic lipid that is a sulfur-containing analog of DLin-MC3-DMA, a helper lipid, cholesterol, and PEG-DMG-encapsulating siRNA. The study verified the successful spray-drying procedure of LNP–siRNA systems while maintaining their integrity and mediating strong gene-silencing efficiency on mRNA and protein levels both in vitro and ex vivo [32].

Overall, the reasons that mRNA vaccines or other forms of therapy are so powerful lie in the many advantages of mRNA therapy technology. (1) Safety: in contrast to DNA, which needs to enter the nucleus, mRNA functions in the cytoplasm, thereby avoiding the risk of its integration into the host genome [33]. (2) High efficiency: appropriate modification regulation and sequence optimization can provide a significant increase in mRNA stability and translation efficiency [34,35,36], and an efficient delivery system has been developed to enable the rapid uptake and cytoplasmic expression of mRNA. (3) Short development period: once the genome sequence of a pathogen is determined, the mRNA that encodes the antigenic protein can be designed [37], and the transcript of the mRNA is produced in vitro without the need for cellular amplification, which significantly increases the speed of production. (4) High flexibility: in the case of a mutation of a virus whereby the original vaccine becomes ineffective, a new mRNA vaccine can be rapidly redesigned and produced based on the new viral sequence. (5) Wide range of applications: mRNA technology is making breakthroughs in the treatment of infectious diseases, genetic diseases, cancer, diabetes, etc., by virtue of its advanced principles and high efficiency in research and development and production [38].

## 5. Future Directions

Despite being so powerful, LPN formulations still have major unsolved issues, including biological barriers to LPN delivery, tissue accumulation, the maintenance of prolonged protein expression, immunological responses, and endosomal escape (extensively reviewed in [39]). LNPs’ surface charge is an additional feature that can be tailored for targeting abilities. Anionic carriers have been utilized for brain therapeutics; in a comparative study, Gabal et al. reported 1.2-fold higher brain targeting efficiency for anionic nanostructured lipid carriers compared with their cationic counterparts [40]. However, anionic particles have limitations due to difficulties faced in nucleic acid packaging and poor transfection efficiency. Tagalakis et al. showed that PEGylation enhances the receptor-mediated transfection efficiency of anionic nanocomplexes. They used cationic targeting peptides as connecting bridges between pDNA and PEGylated anionic liposomes. Unsurprisingly, the newly produced structures displayed more resistance to aggregation in both serum and transfected cells. They also demonstrated increased levels of tissue penetration and broader cellular transfection compared with homologous, non-PEGylated anionic and cationic systems [41]. Anionic integrin-targeted hybrid nanocarriers were also explored for the siRNA treatment of neuroblastoma with reduced systemic and cellular toxicity and minimal clearance by the liver. Anionic receptor-targeted nanocomplexes were found to be specific and efficient as their cationic equivalents. This was also evident in an animal model, since anionic receptor-targeted nanocomplexes transfected tumors in an integrin-mediated fashion and effectively entered tumors with little off-target biodistribution [42].

## 6. Conclusions

mRNA recently emerged as an appealing alternative to treat and prevent diseases ranging from cancer and Alzheimer’s disease to COVID-19 with significant clinical outputs. In vitro-transcribed mRNA has been engineered to mimic the structure of natural mRNA for vaccination, cancer immunotherapy, and protein replacement therapy. In the past few decades, significant progress has been made in unveiling mRNA’s molecular pathways, controlling its translatability and stability, and understanding its evolutionary defense mechanism. However, numerous unsolved structural, biological, and technical difficulties hamper the successful implementation of the systemic delivery of mRNA for safe human consumption. Advances in designing and manufacturing mRNA and selecting innovative delivery vehicles are mandatory to address the unresolved issues and achieve the full potential of mRNA drugs [43].

The current progress in mRNA therapeutics and advancements in designing biomaterials and delivery strategies, including nano- and microscale biodrug delivery systems, may help meet the existing translational challenges of clinical tractability and the prospects of overcoming any challenges related to mRNA [44]. Current efforts in nanomedicine research are seeking to improve LNP and other traditional nanoparticle designs through the use of biomimicry. Biomimetic and bioinspired nanoparticle systems are designed by leveraging solutions from naturally occurring biological structures that have evolved to overcome specific challenges. The manner in which a nanoparticle takes advantage of biomimicry can vary greatly [9].

## Figures and Tables

**Figure 1 ijms-24-02218-f001:**
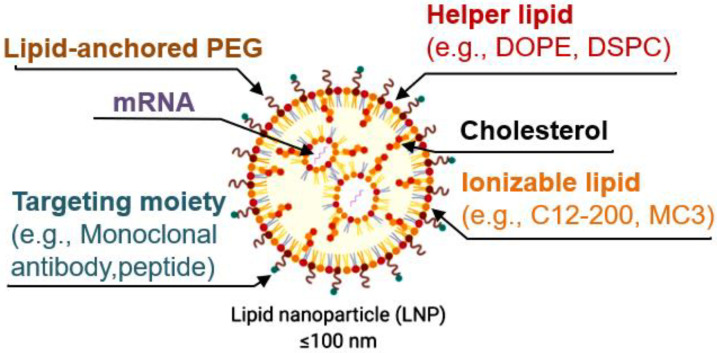
LNP formulation of mRNA. Components are mixed via chaotic mixing via microfluidic device. Interaction of lipid nanoparticle with the lipid bilayer of eukaryotic cells will then occur. Fusion merges the lipids of the nanoparticle with the ones in the cell membrane, ultimately releasing the contents of the nanoparticles inside the cell. Adsorption occurs when the lnanoparticles are attracted to the cell membrane by electrostatic forces, ultimately promoting the release of the cargo inside. Lipid exchange happens when the lipids from the cell membrane and the ones from the lnanoparticle switch. Endocytosis will then occurs when the phagocyte cells engulf the nanoparticles. The inclusion of a targeting moiety—a target cell-specific biomimetic ligand meant to bind an appropriate acceptor on a cell type—is often referred to as nanoparticle surface “functionalization” or “decoration”, and it is meant to improve targeted delivery of the RNA cargo.

**Figure 2 ijms-24-02218-f002:**
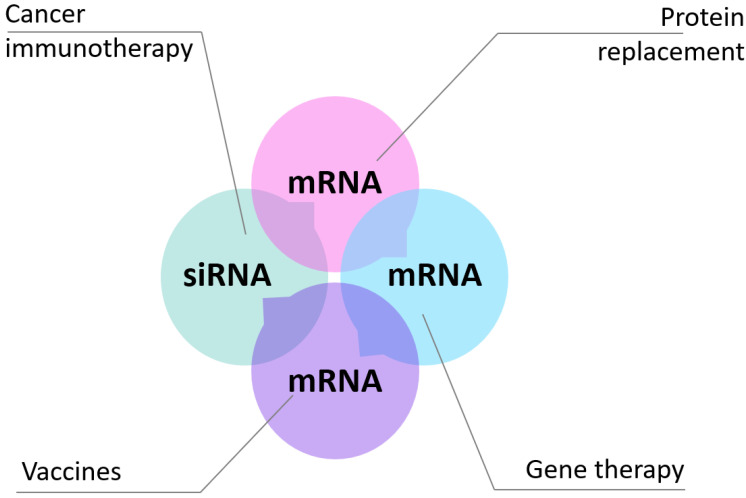
Lipid nanoparticle-mediated delivery of oligonucleotides and mRNA therapeutics. RNA lipid nanoparticle technology can be used to both induce and inhibit gene product expression. In particular, in neoplasia, protein antigen expression can be used for immunotherapy, creating targets for host antitumor immune responses. At the same time, siRNA can be used to inhibit the specific expression of malignant genes.

**Table 1 ijms-24-02218-t001:** Representative list of clinical trials in the United States investigating new mRNA-based interventions.

mRNA Intervention	Type of Infection	Identifier	Phase
mRNA-1189	Epstein–Barr virus infection	NCT05164094	Phase 1
mRNA-1644	HIV infection	NCT05001373	Phase 1
BG505 MD39.3 mRNA	HIV infection	NCT05217641	Phase 1
mRNA-1653	Human metapneumovirus and human parainfluenza infection	NCT04144348	Phase 1
mRNA-1010	Seasonal influenza infection	NCT04956575	Phase 1/2
mRNA-1020 mRNA-1030	Seasonal influenza infection	NCT05333289	Phase 1/2
mRNA-1893	Zika virus infection	NCT04917861	Phase 2
mRNA-1647	Cytomegalovirus infection	NCT05085366	Phase 3
mRNA-1345	Respiratory syncytial virus infection	NCT05330975	Phase 3

## Data Availability

Not applicable.

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
