# Peer review of "Biodrug Delivery Systems: Do mRNA Lipid Nanoparticles Come of Age?"

_ijms, 2023, doi:10.3390/ijms24032218_

Round 1

Reviewer 1 Report

This study focuses on the challenges and applications of micro and nanotechnologies, especially mRNA lipid nanoparticles, in biodrug delivery systems. However, I believe the quality of this work as well as writing is not so good to meet the criteria of International Journal of Molecular Sciences. A few comments are listed below:

1.  The manuscript is suggested to be carefully checked before submission to avoid typos and grammatical issues.   2.  The author needs to rewrite the abstract part in the text. I suggest to highlight the center of discussion in this paper based on the introduction of several nano-delivery systems, i.e., mRNA lipid nano-delivery systems.   3.  Figure2 is not clear enough to visually express several therapeutic approaches of lipid nanoparticle-mediated oligonucleotides and mRNA.   4.   In the "Applications" section, the author cites a significant number of references. However, some literatures are not direct examples to show their contributions to this field. I would suggest that the author adds some more citations for comparisons.

Reviewer 2 Report

As an appealing alternative to treat and prevent diseases ranging from cancer and Alzheimer's disease to COVID-19, mRNA demonstrated a significant clinical effect. Nanotechnology facilitates the successful implementation of systemic delivery of mRNA for safer human consumption. In this manuscript, Matteo Puccetti et al. provide an overview of current mRNA therapeutic applications and discuss key biological barriers to delivery and recent advances in the development of nonviral systems. The relevant challenges that LNPs face in achieving cost-effective and widespread clinical implementation when delivering mRNA were discussed adequately. However, this manuscript has a lot of room for improvement in content and figures. Thus it was recommended for publication after major revision.

1.      In the abstract, the content is too broad and does not reflect the focus of this review. The mRNA lipid nanoparticles are the main subject of this article, not all nanotechnologies, and the authors should have pointed that out to prevent misleading readers.

2.      In the introduction, the statement, “Nanocarriers are achieved from numerous inorganic and organic compounds, such as polymers, dendrimers, lipids, proteins, antibodies, peptides, cells, metals and metal oxides (e.g., iron oxide, silver and gold), non-metal oxides (silica), quantum dots, carbon nanotubes and mesoporous materials”, is not stringent enough. Some materials listed in this sentence are not in the common level of classification, which have including or included relationships with each other.

3.      In the introduction, the authors should add the necessary introduction to mRNA and clarify whether the saRNA mentioned in the manuscript is self-amplifying mRNA or self-amplifying RNA.

4.      Some noun abbreviations need to indicate their full names when they appear for the first time in the text. The labeling of abbreviations and full names in this manuscript is confusing, such as saRNA, in vitro transcription (IVT), self-amplifying mRNA (SAM) and so on.

5.      The punctuation after “e.g.” in the brackets in Figure 1 is not uniform. In addition, the directional text description in Figure 1 can be adjusted to make it neat and beautiful.

6.      The description of Figure 2 in section 2 “mRNA non-viral delivery approaches” does not match the content of the figure itself. Specifically, The classification for therapeutic applications of mRNAs in section 2 are protein replacement strategies, vaccine production and gene editing, and cancer immunotherapy is classified as a type of vaccine. But Figure 2 juxtaposes the above four therapeutic applications of mRNAs, which is obviously unreasonable. What’s more, the composition of Figure 2 makes it difficult for readers to see the complete content and the legend is too simple to reflect the substance.

7.      In Table 1, what does the serial number in front of some mRNA intervention in the first column mean? It seems to break the order of the entire table.

8.      In section 4 “Applications”, the authors introduced the research on the genes of Alzheimer's disease. However, this content seems to have nothing to do with the LNP delivery of mRNA. Please explain the reason for writing this content.

9.      When discussing the current progress of mRNA therapeutics and advancements in designing biomaterials and delivery strategies, the authors put forward the challenges and deficiencies in this field. If the authors can propose suggestions and prospects for these deficiencies, then the reference value of this article will be greatly improved.

10.  Some references are missing page numbers or document numbers.

Reviewer 3 Report

In the present manuscript, a description of present mRNA therapeutics and barriers to deliver the therapeutics and the recent advances in the development using non-viral systems have been explained. My specific comments were provided below:

1.     The sentence in 26th line in the manuscript can be shifted to one line above to make the story continuous.

2.     More examples regarding the mRNA based LNP formulations can be explained in the text to show their effectiveness and importance.

3.     The title of the article can be simplified in order to make it more understanding to the readers.

4.     Some of the future prospects regarding overcoming the drawbacks associated with the use of lipid nanoparticles can be addressed in the manuscript.
